



# Brief communication: Do 1.0°C, 1.5°C or 2.0°C matter for the future evolution of Alpine glaciers?

Loris Compagno[1,2], Sarah Eggs[1,2], Matthias Huss[1,2,3], Harry Zekollari[5,4,1,2], and Daniel Farinotti[1,2]

[1]Laboratory of Hydraulics, Hydrology and Glaciology (VAW), ETH Zurich, Zurich, Switzerland.
[2]Swiss Federal Institute for Forest, Snow and Landscape Research (WSL), Birmensdorf, Switzerland.
[3]Department of Geosciences, University of Fribourg, Fribourg, Switzerland.
[4]Laboratoire de Glaciologie, Université libre de Bruxelles, Belgium.
[5]Department of Geoscience and Remote Sensing, Delft University of Technology, Netherlands.

**Correspondence:** Loris Compagno <compagno@vaw.baug.ethz.ch>

**Abstract.** With the Paris Agreement, the urgency of limiting ongoing anthropogenic climate change has been recognized. More recent discussions have focused on the difference of limiting the increase in global average temperatures below 1.0, 1.5, or 2.0°C compared to pre-industrial levels. Here, we assess the impacts that such different scenarios would have on both the future evolution of glaciers in the European Alps and the water resources they provide. Our results show that the different
temperature targets have important implications for the changes predicted until 2100, and that glaciers might start recovering after the end of the 21st century.

## 1 Introduction

Climate change is one of the largest challenges that society will need to face during this century. There is overwhelming consensus that limiting the increase of global average temperatures below a certain threshold is essential if adverse effects
are to be avoided (IPCC, 2013). Put forward in 2015 and adopted by 196 parties, the Paris Agreement concurred that such a threshold is *"well below 2°C above pre-industrial levels"*, and that efforts should be pursued to *"limit the temperature increase to 1.5°C"* (UN, 2015). Even under these ambitious targets an important environmental change is to occur. The Special Report on Global Warming of 1.5°C (IPCC, 2018), for example, focused on the effects that would result if the 1.5°C target was to be met, and highlighted that even under such conditions, important and mostly negative changes would occur. Five years after
the Agreement, there is compelling evidence that any reduction of anthropogenic climate change will pay off in the longer term, both from an ecological (IPBES, 2019) and economical (WEF, 2020) perspective. Discussions in preparation for the renewed pledges by signatory parties have thus focused on the differences between scenarios close to the original targets. For the upcoming 26[th] United Nations Climate Change Conference of the Parties, for example, emphasis is put onto separating the impacts occurring under scenarios of 1.0, 1.5, and 2.0°C of warming above pre-industrial levels. In this brief communication,
we focus on the differing impacts that such scenarios would have on glaciers and related water resources in the European Alps. Across the world, glaciers are amongst the most prominent indicators for climatic change, as they show an integrated response of climate forcing over decades to centuries (e.g. Roe et al., 2016). Previous work estimated that every kg of additionally



emitted $CO_2$ would result in a long-term glacier mass loss of ca. 16 kg (Marzeion et al., 2018) – with important implications
for the corresponding landscapes, downstream ecosystems, and water supplies (IPCC, 2018). Whilst future projections for the
glacier evolution of the European Alps already exist (e.g. Huss and Hock, 2015; Zekollari et al., 2019), targeted information
on policy-relevant climate targets is difficult to tease out. Here we do so by re-running the GloGEMflow model (Zekollari
et al., 2019) with updated climate projections from the 6ᵗʰ phase of the Coupled Model Intercomparison Project (CMIP6). In
particular, we focus on Shared Socioeconomic Pathways (SSPs) that would result in global mean temperature changes of +1.0,
+1.5 and +2.0°C by 2100 compared to pre-industrial conditions (1850-1900), and provide projections for both future glacier
geometries and glacier runoff until the end of the century. In an explorative analysis, we also consider projections until the year
2300, albeit acknowledging that large uncertainties exist when doing so.

## 2   Data and Methods

We simulate the evolution of all 3927 glaciers in the Alps independently based on (i) outlines provided by the Randolph Glacier
Inventory version 6 (RGI 6.0) (RGI Consortium, 2017) and (ii) the consensus ice thickness estimate (Farinotti et al., 2019).
35       Glacier evolution is modelled with the combined mass-balance ice-flow model GloGEMflow (Huss and Hock, 2015; Zekol-
lari et al., 2019). Accumulation is computed based on precipitation and temperature, with a linear transition between solid
and liquid precipitation occurring for temperatures between 0.5°C and 2.5°C. Ablation is calculated with a temperature-index
model (Hock, 2003), and refreezing is determined based on heat conduction. The mass balance module runs at a monthly reso-
lution, including a parameterization for day-to-day temperature variability. It is calibrated to reproduce glacier specific-geodetic
volume changes (Zemp et al., 2019) spanning approximatively the period 1959– 2013 and covering 57 % of the Alpine glacier
area. For more details on the mass balance module and its calibration, refer to Huss and Hock (2015).
       For all glaciers with a length >1 km in RGI 6.0 (i.e. 795 glaciers, accounting for 95 % of the Alpine glacier volume), we
compute ice flow based on the shallow ice approximation. The computations follow Zekollari et al. (2019), including the
iterative initialisation procedure used to generate a transient glacier geometry for the RGI inventory year (i.e. 2003 for the
majority of Alpine glaciers). For all glaciers with length <1 km, instead, glacier evolution is modelled with an elevation-
dependent parameterization that has shown to be in good agreement with higher-order ice-flow modeling results (Huss et al.,
2010). For more information on GloGEMflow's glacier evolution modelling, please refer to Zekollari et al. (2019).
       For each glacier, water runoff is calculated at a monthly resolution by summing rain and melt (from ice, snow and firn) and
by subtracting refreezing. Throughout the simulations, runoff is computed for the area comprised within the RGI 6.0 outlines,
implying that, after glacier retreat, runoff contributions from rain and snow melt are still accounted for (in line with Huss and
Hock, 2018).
       For the past (1950-2020), GloGEMflow is forced by monthly 2-m air temperature and precipitation taken from the ensemble
daily gridded observational dataset (E-OBS v.21.0) at 0.1° horizontal resolution (Cornes et al., 2018). For the future (2020-
2100), we rely on 128 CMIP6 global circulation model (GCM) members. Consistence between past and future datasets is
ensured by applying the de-biasing procedure proposed by Huss and Hock (2015). In a nutshell, the procedure uses a set of





additive and multiplicative correction factors to adjust both the long-term mean and the long-term variability of the coarse-resolution GCMs to the level of the high-resolution E-OBS data.

To evaluate model performance, we compare modelled mass balances with observations provided by the World Glacier Monitoring Service (WGMS, 2020) for 72 glaciers. For glacier-wide annual mass balance, we find a bias of 0.07 m w.e. a$^{-1}$,

a Root Mean Square Error (RMSE) of 1.07 m w.e. a$^{-1}$, and a squared correlation coefficient (r$^2$) of 0.29 (see Fig. S1). Observations aggregated to elevation bands yield a bias of $-0.01$ m w.e. a$^{-1}$, a RMSE of 1.47 m w.e. a$^{-1}$, and a r$^2$ of 0.52. For glacier-wide winter balance, instead, a bias of 0.20 m w.e. a$^{-1}$, a RMSE of 0.74 m w.e. a$^{-1}$, and a r$^2$ of 0.26 is found (see Fig. S2). Albeit correlation coefficients are rather low, the small biases provide confidence in the regional results. The latter is also confirmed by the comparison of the total ice volume change 2000-2020 in Switzerland (comprising almost 60% of the Alpine

glacier volume at present): estimated as 23.6 km$^3$ based on observational series (Grab et al., submitted), it is very close to our modelled loss of 24.9 km$^3$. Similarly, the modelled rate of glacier area loss (1.4 % a$^{-1}$ for the period 2003–2016) is in very good agreement with the rate of 1.3 % a$^{-1}$ derived by Paul et al. (2020) using Landsat images.

For our main analysis, finally, we only select those GCM members that yield a 21st century warming of $+1.0$, $+1.5$ and $+2.0°$C compared to pre-industrial. This selection is performed by computing the mean global 2-m air temperature change

over both land and ocean between the period 2071-2100 and pre-industrial (1850-1900) levels, and by selecting members that are within a given range of temperature increase: $+1 \pm 0.25°$C (3 model members), $+1.5 \pm 0.25°$C (11 model members), and $+2 \pm 0.25°$C (14 model members). The selected members correspond to SSPs that assume low emissions, i.e. SSP119, SSP126 and SSP245, and are used to force GloGEMflow for the period 2020-2100. To determine the corresponding warming in the European Alps (see Fig. 1 and S3), we extract all GCM grid cells that are within latitudes of 42.5°N - 49.5°N, longitudes of

4°E - 16°E, and at elevations above 500 m.

## 3  Results and Discussion

The GCMs selected within the three temperature change targets (i.e. $+1.0$, $+1.5$, and $+2.0°$C) show a global mean warming between 1850-1900 and 2071-2100 of $+1.00 \pm 0.11$, $+1.49 \pm 0.17$, and $+2.07 \pm 0.18°$C. For the European Alps, this corresponds to a temperature change of $+0.98 \pm 0.56$, $+1.80 \pm 0.72$ and $+2.51 \pm 0.73°$C, respectively (Fig. 1a). During the summer

months (JJA), the temperature increase in the Alps is even stronger, and is of $0.96 \pm 0.88$, $2.09 \pm 1.24$, $2.81 \pm 1.23°$C, respectively. These differences agree well with previous assessments (Vautard et al., 2014), and are attributed to stronger warming over land than over ocean, to enhanced warming at high latitudes as compared to low latitudes, and to amplified warming at higher elevations (Vautard et al., 2014; IPCC, 2013). For a global warming of $+1.0$, $+1.5$, and $+2$ °C, annual precipitation sums by 2071-2100 are projected to increase by $18 \pm 8$, $29 \pm 9$, and $42 \pm 10$ mm a$^{-1}$, respectively, up from 1141 mm a$^{-1}$ in

the global scale, pre-industrial baseline (Fig. 1b). For the Alps, the annual precipitation increase is anticipated to be of $37 \pm 94$, $29 \pm 135$, and $46 \pm 145$ mm a$^{-1}$, respectively (the pre-industrial baseline is 1210 mm a$^{-1}$), whilst precipitation changes over the winter season are slightly amplified compared to the annual signal.





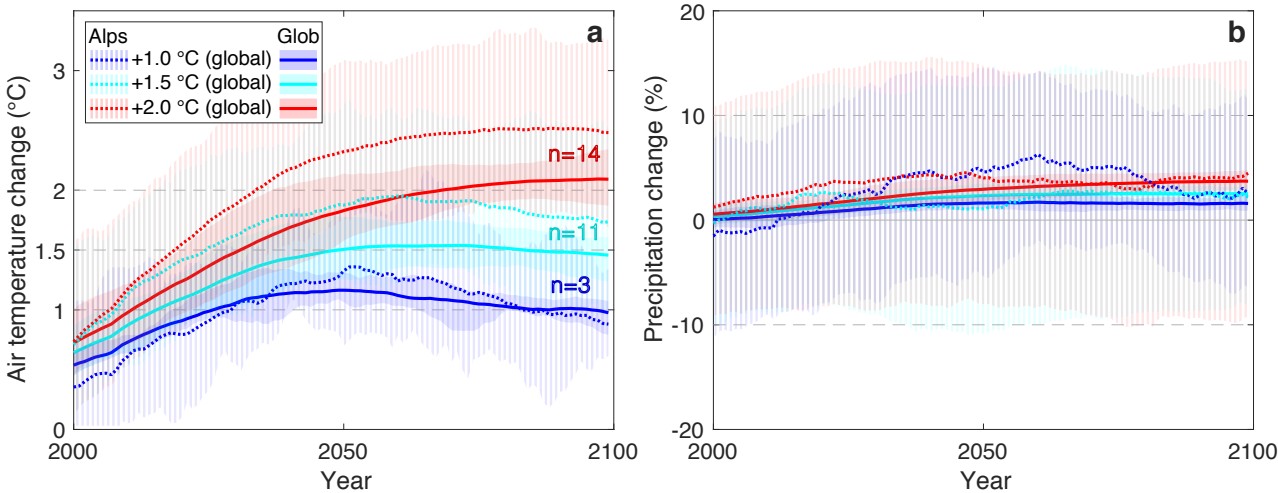

**Figure 1.** Evolution of annual global and Alpine (**a**) average air temperature and (**b**) precipitation total compared to the pre-industrial (1850-1900) baseline. Series are smoothed with a 30-year running mean. The coloured bands show the standard deviation of the projection given by individual GCM members. $n$ is the considered numbers of GCM members.

In all three considered warming scenarios, glaciers will be drastically affected. By 2100, glaciers in the European Alps are projected to lose $44 \pm 21\%$ (+1.0°C), $68 \pm 12\%$ (+1.5°C), and $81 \pm 8\%$ (+2.0°C) of their 2020 ice volume (see Fig. 2a).
The evolution of the glacier area will be on a par ( Fig. 2b). By 2100, Alpine glaciers are projected to lose $47 \pm 16\%$ (+1.0°C), $65 \pm 10\%$ (+1.5.°C), and $77 \pm 8\%$ (+2.0°C) of their 2020 glacierized area (Fig. 2b).

Figure 3a-c shows the regional distribution of the modelled glacier evolution. By 2100, only $1,484 \pm 324$ (+1.0°C), $1,033 \pm 253$ (+1.5°C) and $724 \pm 201$ (+2.0°C) of today's roughly 4,000 glaciers are anticipated to remain (volume $>0$). In the three scenarios, Grosser Aletschgletscher (Fig. 3d), the largest glacier of the Alps, might lose as much as $45 \pm 12\%$, $64 \pm 11\%$, and
$77 \pm 6\%$ of its 2020 volume, respectively. Rhonegletscher (Fig. 3e), one of the most studied and visited glaciers, is projected to lose $55 \pm 24\%$, $78 \pm 11\%$, and $91 \pm 7\%$ of its 2020 volume by 2100. Values for other glaciers are found in Table S1. Our results demonstrate strong sensitivity of glacier response to even small levels of global atmospheric warming, such as +1.0°C. Most importantly, however, our results show significant differences between the future glacier evolution projected for the three warming scenarios, indicating that any effort to further limit warming will have important effects on glaciers and , thus, future
Alpine environment.

Future effects notably include the water availability from presently glacierized areas. Our results indicate that, on a yearly basis, glacier runoff for the period 2080-2100 might decrease by as much as $25 \pm 6\%$ (+1.0°C), $32 \pm 8\%$ (+1.5°C), and $36 \pm 10\%$ (+2.0°C) compared to 2000-2020 levels (Fig. 2c). The decrease follows above-average glacier melt during the last decades, thus exacerbating the effect that decreasing ice volumes have on runoff contributions (Huss and Hock, 2018). Our
model results also show a transition in the timing of the maximum glacier discharge during the year: whilst it occurred in August historically (2000-2020), it is expected to occur in July (+1.0 and +1.5°C scenarios) or even June-July (+2.0°C) by





**Figure 2.** Modelled evolution of total glacier (**a**) volume, (**b**) area, (**c**) annual glacier runoff, and (**d**) monthly glacier runoff of the European Alps. Time series in **c** are smoothed with a 20-year running mean. In all panels, the thick line represents the mean and the transparent band corresponds to one standard deviation of the results obtained by forcing GloGEMflow with the selected GCM members. The numbers or GCM members is given ($n$).

2080-2100 (Fig. 2d). Finally, our results anticipate the magnitude of the runoff peaks to be $36 \pm 15\%$ (+1.0°C), $44 \pm 14\%$ (+1.5°C), and $55 \pm 15\%$ (+2.0°C) lower by 2080-2100 on average than they were in 2000-2020. This shift in the magnitude and timing of glacier runoff is well documented (e.g. IPCC, 2013), is a direct consequence of both the reduced glacier volumes

and the increased temperatures (which cause snow melt to occur earlier in the year), and will have important consequence for water availability (Huss and Hock, 2018), particularly during dry summer months. For the densely populated European Alps and the areas downstream, such changes will not only impact ecosystems but also a number of water users, including agriculture or hydropower companies for instance (Beniston et al., 2018).





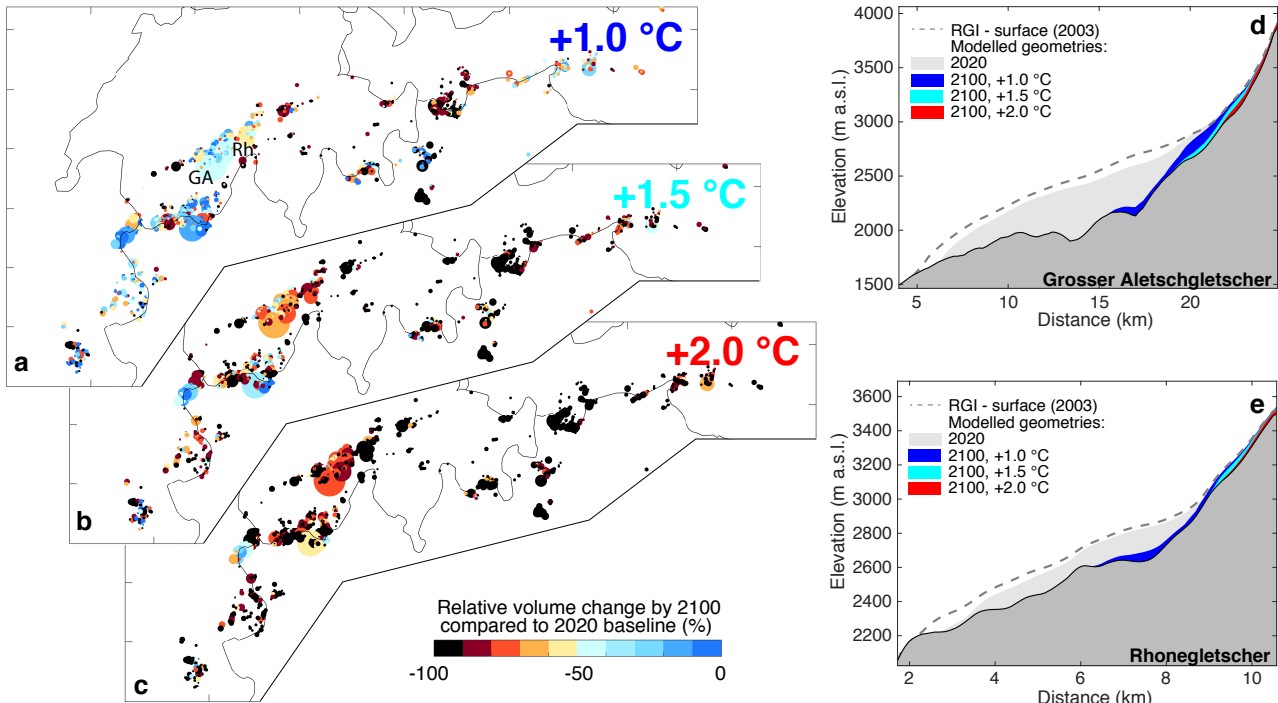

**Figure 3.** Relative change in glacier volume between 2020 and 2100 for scenarios of (**a**) +1.0 °C, (**b**) +1.5 °C and (**c**) +2.0°C of global temperature increase. Each dot represents one glacier, the dot's size being proportional to the glacier area as given by the Randolph Glacier Inventory. The labels "GA" and "Rh" identify "Grosser Aletschgletscher" and "Rhonegletscher", respectively. The modelled geometries of these two glaciers are shown in panels (**d** and **e**) at inventory date (dashed), for the year 2020 (grey), and for the year 2100 under the three selected scenarios (colours).

In line with the bulk of the literature, the above results all refer to the time horizon until 2100. To gain insights on glacier
evolution beyond this horizon, we re-run GloGEMflow with three GCM members that provide climate data until 2300 and that project mean global temperature changes below +2.0°C for that time (see Fig. S3). The three members all come from SSP126 but stem from three different GCMs (see Fig. S3).

These results show that, owing to slow lowering of air temperatures and enhanced precipitation after 2100, slow glacier recovery might happen (Fig. S3). While glacier volume losses of 80-85% are calculated for 2100, the experiment projects
Alpine glaciers to re-gain a total volume that is between 47% and 72 % of the 2020 level by 2300. Although this result is only based on three GCM members and is thus very uncertain, it suggests that considering projections beyond 2100 might change the perception of an irreversible trend. Increasing the number of GCM members that consider such longer-term horizons and having different research groups performing similar analyses would help verifying the robustness of this preliminary finding.

## 4 Conclusions

Whilst there is overwhelming consensus that decisive acting has to be taken to limit unwanted consequences of ongoing climatic change, the debate around which climatic target to pursue is all but settled. In this contribution, we quantified the impact that a global temperature change of +1.0°, +1,5° or +2.0°C compared to pre-industrial levels would have on glaciers of the European Alps. We showed that this would correspond to an Alpine warming slightly above global levels and to a moderate increase in precipitation. Under such conditions, Alpine glaciers are projected to lose $44 \pm 21\,\%$, $68 \pm 12\,\%$, and $81 \pm 8\,\%$ of

their 2020 ice volume by 2100, indicating that even in the most moderate scenario (+1.0°C), about half of the present-day glacier volume would be lost by the end of the century. Preliminary results based on a strong mitigation scenario (SSP126) and running until 2300 indicate that slow recovery might happen after that, emphasizing the interest in considering projections that reach beyond the 21st century. The changes in glacier volume will strongly impact the water yield from presently glacierized surface, with 2080-2100 annual average runoff decreasing by $25 \pm 6\,\%$ (for a global warming of +1.0°C), $32 \pm 8\,\%$, (+1.5°C)

and $36 \pm 10\,\%$ (+2.0°C) compared to 2000-2020 levels. Changes in monthly runoff – anticipated to occur 1 to 2 months earlier by the end of the century – will be even more pronounced, with August runoff reductions of $36 \pm 15\,\%$ , $44 \pm 14\,\%$, and $55 \pm 15\,\%$, respectively. For glacierized catchments and the areas downstream, these changes in water availability can be expected to have important consequences ranging from the functioning of ecosystems to the generation of hydro-electricity. Even if our preliminary results based on a strong mitigation scenario indicate that slow glacier recovery might happen after the

21ˢᵗ century, we hope that the results presented here will contribute in making the point that, when it comes to global climate, every half-degree counts.

*Code availability.* Code and data availability upon request

*Author contributions.* LC and SE performed the numerical modelling with support from MH, HZ and DF. The original code was developed by MH (mass balance component) and by HZ (ice flow component). DF, LC, SE conceived the study. SE downloaded all CMIP6 GCMs,

with support of LC, MH and DF. LC and DF wrote the manuscript and produced the figures, with contributions from all other co-authors.

*Competing interests.* There are no competing interests.

*Acknowledgements.* We are grateful to the Swiss National Science Foundation, and for the funding provided for project Nr. 184634 in particular. HZ acknowledges the funding from a Marie Skłodowska-Curie Individual Fellowship (Grant 799904). We thank the RGI consortium for the global glacier inventory data. We acknowledge the ECA&D project for the E-OBS dataset, and CMIP for the GCM outputs. We also

thank the WGMS for providing mass balance and length change measurements.





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
