# Peer review of "Brief communication: Do 1.0°C, 1.5°C or 2.0°C matter for the future evolution of Alpine glaciers?"

_The Cryosphere, 2021_

## Referee Comment (RC1)

Review of

**Brief communication: Do 1.0°C, 1.5°C or 2.0°C matter for the future evolution of Alpine glaciers?**

Journal: The Cryosphere
Date: 2021-03-15

Summary:

This paper aims to assess the influence of mitigating global climate warming less than 2.0°C on Alpine glacier evolution and its related water resources. This goal is achieved by modelling the glacier surface mass-balance and ice dynamic using the glacier model GloCEMflow, forced by temperature and precipitation extracted from CMIP6 experiments (global climate models) and calibrated against *in situ* mass balance observations. Results can be nicely summarized by this words: **for Alps glacier changes and their consequences, every half-degree counts**. In order to maintain a certain amount of glacier ice volume in the European Alps (19% +- 8% for 2°C of global warming in 2100 compared to pre-industrial values and 56% +- 21% for 1°C), important mitigation strategies are needed. Furthermore, annual average runoff over glacierized catchments will decrease between 25% +- 6% and 36% +- 10% depending on the global warming target, and peak seasonal water will be advanced by 1 or 2 months.

General remark:

Overall, the paper is well presented, English is clear and the work done is robust, with a high scientific rigour and with a clear and concise message. Data and model are clearly presented and the methodology seems to be the good one to reach the study goal. I thus think the article deserve publication, especially for its importance and impact on policy maker debates about their procrastination to act for maintaining global warming under certain temperatures (The Paris Agreement in 2015).

One downside is the moderate originality of the study. Indeed, glacier volume decrease for different global warming targets has already been studied at global scales (e.g. Marzeion et al., 2018, or SROCC report,

2019), thus including the European Alps, and this study does not improve too much the expected regional results but confirmed them (see also Zekollari et al., 2019). However, that's the first time that CMIP6 experiment outputs are used to drive the glacier model.

Finally, we could also imagine other interesting studies within the framework and method used here (even if that's out the scope of Paris Agreement): how the European glaciers will evolve for higher global warming targets ? does the half degree between 1°C and 1.5°C count as the same manner as the half degree between 3°C and 3.5°C ? how many years do we have until actual committed glacier mass loss become similar to the different global climate warming targets ? what are the spatial patterns of future climatology over the European Alps glaciers and at finer scales, and how those patterns will affect glacier evolution ?

Specific comments:
- **line 8 to 31:** I think references to SROCC, 2019, and GlacierMIP2 papers are missing. In addition, one or two sentences could be useful to explain why you are using this model instead of OGGM (Maussion et al., 2019) or PyGEM (Rounce et al., 2020) for example.
- **line 48 to 51**: for water runoff calculation, you do not take into account rain which is outside glacier outlines but at higher elevation than the glacier front (for example other part of the mountain catchment area which are not cover by ice), neither snow melt outside the glacier (but likewise at higher elevation than the glacier terminus). How this will affect the runoff estimates you are calculating ?
- **line 52 to 58**: there is a certainly a lack in the representation of spatial variations of temperature and precipitation over the mountainous Alpine region with the GCMs used. How this will affect the results ? Why not having used RCMs do downscale the data ? Or used other regional climate forcing such as EUROCORDEX ensemble ?
- **line 63**: how can you explain the relatively high RMSE and low square correlation coefficient whereas the bias is low when comparing glacier-wide mass-balance between GloCEMflow simulations and the 72 WGMS observations ?

- **line 58 to 67**: evolution of glacier volume change can also be compared with Zekollari et al., 2019, and Marzeion et al., 2020 (Partitioning the uncertainty of ensemble projections of global mass change), even if both studies do not target specifically 1.0°C, 1.5°C and 2.0°C global climate changes, and that the second study is global. Comparison with the study of Zemp et al., 2019: they found 2 092 $km^2$ of glacier area in the region 11 (European Alps), multiply by 0.97 . $10^{-3}$ km.we.yr$^{-1}$ (0.87 . $10^{-3}$ / 0.9 ice density = 0.97 . $10^{-3}$), it gives 2.029 $km^3$.we.yr$^{-1}$ of mass loss per year. Thus, over 20 years, it results to 40.585 $km^3$ glacier mass loss and finally only over Switzerland (60% of the Alps ice volume), it gives 24.35 $km^3$ glacier mass lost during the last 20 years which is also very close to what you found ! Are my first order calculation right ?
- **line 88 to 91**: you do not discuss that large part of glacier mass in the Alps is already committed to melt because actual temperatures are already largely higher than pre-industrial values ?
- **line 92 to 93**: authors explain the different remaining glaciers in 2100 for different global climate warmings. I wonder how many glacier are committed to disappear in the European Alps under actual climate conditions (that is close to 1.0°C above pre-industrial values) ?
- **line 97**: do not forget that global average warming of 1.0°C does not result regionally of 1.0°C exactly of warming and thus glaciers can experienced large differences.
- **line 101 to 113**: is it feasible and physically consistent/interesting to go to daily resolution for runoff calculation ?
- **line 122**: I think authors could be more incisive about the irreversible glacier trend. Physically, glaciers growths and retreats are totally consistent and reversible, thus that's not surprising that if global temperature starts to stabilize and reduce after 2100 horizon, glaciers start to grow again at regional scale.
- **line 141**: I am surprised that what seems to me the key message, i.e. "every half-degree count", is not more highlighting in the paper (in the abstract for example).

- **figure 1**: to be consistent between panel a) and b), the vertical axis should be either in unit or in unit change (%) for both panels.
- **figure 1 and 2**: is it consistent to choose a moving average window of 30 years for climate averaging and 20 years for runoff averaging ?
- **figure 2:** why there is no peak water in the curves for annual glacier runoff (panel c) ? How does it compare to other studies ? Does it mean that the peak water is already reached for glaciers in the European Alps ?

- **supplementary material, table 1**: read again the legend which is I think not clear. Probably remove "given as Area ??" and please explicit "w.r.t".

---

## Author Response (AR1)

**Author's response to the comments received for tc-2021-31**

The following pages contain a point-by-point reply to the comments provided by the two referees that reviewed our first submission (TC-2021-31)

 Each of the referee's comment (**RC**) is numbered. If a comment contained several points, we numbered them, and address them individually in our author replies (**AR**).

**1.  REFEREE'S COMMENT**

General remark:
**[RC 1.01]** Overall, the paper is well presented, English is clear and the work done is robust, with a high scientific rigour and with a clear and concise message. Data and model are clearly presented and the methodology seems to be the good one to reach the study goal. I thus think the article deserve publication, especially for its importance and impact on policy maker debates about their procrastination to act for maintaining global warming under certain temperatures (The Paris Agreement in 2015).

**[AR 1.01]** We thank the reviewer for the very positive feedback.

**[RC 1.02]** One downside is the moderate originality of the study. Indeed, glacier volume decrease for different global warming targets has already been studied at global scales (e.g. Marzeion et al., 2018, or SROCC report, 2019), thus including the European Alps, and this study does not improve too much the expected regional results but confirmed them (see also Zekollari et al., 2019). However, that's the first time that CMIP6 experiment outputs are used to drive the glacier model.

**[AR 1.02]** We partially agree with this comment, and appreciate that the use of the new CMIP6 results was noted. Our goal was to clearly focus on the difference that 0.5°C in global temperatures can make at the regional scale. We believe this message to be very important, e.g. for the upcoming 26th United Nations Conference of Parties and for policy makers. Without wanting to drift towards activism, we would like to provide additional evidence for the urgency of taking action against any further warming. We slightly amended ll.26-29 to acknowledge that previous works on the topic exist, though:

 *'Whilst future projections for the glacier evolution of the European Alps already exist under different representative concentrations pathways (RCPs) (e.g. Zekollari et al., 2019, Marzeion et al., 2020), targeted information on policy-relevant climate targets (like the difference between 1.5 and 2.0°C IPCC, 2018) is difficult to identify.'*

**[RC 1.03]** Finally, we could also imagine other interesting studies within the framework and method used here (even if that's out the scope of Paris Agreement): how the European glaciers will evolve for higher global warming targets? does the half degree between 1°C and 1.5°C count as the same manner as the half degree between 3°C and 3.5°C ? how

many years do we have until actual committed glacier mass loss become similar to the different global climate warming targets? what are the spatial patterns of future climatology over the European Alps glaciers and at finer scales, and how those patterns will affect glacier evolution?

[AR 1.03] We absolutely agree with the reviewer that a number of additional analyses could be conducted with the same framework. We had discussed these possibilities within our team of co-authors but finally agreed that the limited space offered by a brief communication would make it very difficult to address all these questions. We also argue that a relatively simple message would be beneficial for the policy-related impact that we envision this paper to have. We hope that our reasoning is understandable, and that it will find the reviewer's support.

Specific comments:

[RC 1.04] line 8 to 31: I think references to SROCC, 2019, and GlacierMIP2 papers are missing. In addition, one or two sentences could be useful to explain why you are using this model instead of OGGM (Maussion et al., 2019) or PyGEM (Rounce et al., 2020) for example.

[AR 1.04] We agree that SROCC (2019) and Marzeion et al. (2020) (i.e. GlacierMIP2) are important references in this context and have included them. As for what the model choice is concerned, note that GloGEMflow is our in-house glacier model and thus clearly our primary choice. Although OGGM or PyGEM would certainly be valid alternatives, inter-comparison projects such as the now-cited GlacierMIP2 do not provide any evidence that other models might be better suited than ours. We now briefly mention this reasoning with the following sentence (ll.28-32):
'Here we analyse glacier evolution under low global warming scenarios by re-running our in-house mass-balance ice-flow model GloGEMflow (Huss and Hock, 2015; Zekollari et al., 2019), that showed robust performance in recent model intercomparison projects (e.g. Marzeion et al., 2020), with updated climate projections from the 6th phase of the Coupled Model Intercomparison Project (CMIP6). '

[RC 1.05] line 48 to 51: for water runoff calculation, you do not take into account rain which is outside glacier outlines but at higher elevation than the glacier front (for example other part of the mountain
catchment area which are not cover by ice), neither snow melt outside the glacier (but likewise at higher elevation than the glacier terminus). How this will affect the runoff estimates you are calculating ?

[AR 1.05] We are not entirely sure to understand the reviewer's comment. The contents of the comment are correct (i.e. it is true that the mentioned contributions are not accounted for) but this is exactly how we define our focus at l. 53-54: 'runoff is computed for the area comprised within the RGI 6.0 outlines, implying that, after glacier retreat, runoff contributions from rain and snow melt are still accounted for.' Throughout the simulations, runoff refers to the initial glacier surface, in line with e.g. the work by Huss and Hock (2018). That said, please note that the runoff contribution from rain or snow melt coming from areas that become ice-free during the simulated time period is indeed accounted for. This is to avoid the runoff contribution to carry the signal of a shrinking glacier area.

**[RC 1.06]** line 52 to 58: there is a certainly a lack in the representation of spatial variations of temperature and precipitation over the mountainous Alpine region with the GCMs used. How this will affect the results ? Why not having used RCMs do downscale the data ? Or used other regional climate forcing such as EUROCORDEX ensemble ?

**[AR 1.06]** We used GCMs and not RCMs to force our model because no RCM model runs exist for CMIP6 at the moment. Whilst it would be beyond the scope of a brief communication to perform an EURO-CORDEX-type of RCM-downscaling for all available CMIP6 GCM runs, we note that the GCM results are downscaled to the E-OBS dataset (described at ll.62-63), and that we don't expect this choice to have a large impact on our results. Indeed, the recent analysis by Compagno et al. (2021) showed that – provided suitable model calibration data – the impact of using either GCMs or RCMs as model forcing is very limited. We now acknowledge this point with the following sentence:

ll. 58-60: *'Here we use GCMs, Since CMIP6 GCM-results have not yet been downscaled by regional climate models, and since the spatial resolution of the climate data was recently shown to only marginally affect model results if suitable model calibration at the glacier-specific scale is ensured (Compagno and others, 2021).'*

**[RC 1.07]** line 63: how can you explain the relatively high RMSE and low square correlation coefficient whereas the bias is low when comparing glacier-wide mass-balance between GloGEMflow simulations and the 72 WGMS observations ?

**[AR 1.07]** The values indicate that individual results are affected by some degree of scatter but that there is no general tendency to either over-estimate or under-estimate the results as the general mass change signal is given by observed geodetic mass balances. We clarified this with the following sentence:

ll. 70-71: *'Albeit correlation coefficients are rather low, the small biases provide confidence in the regional results, which neither indicate over- nor underestimation.'*

**[RC 1.08]** I line 58 to 67: evolution of glacier volume change can also be compared with Zekollari et al., 2019, and Marzeion et al., 2020 (Partitioning the uncertainty of ensemble projections of global mass change), even if both studies do not target specifically 1.0°C, 1.5°C and 2.0°C global climate changes, and that the second study is global. Comparison with the study of Zemp et al., 2019: they found 2 092 km2 of glacier area in the region 11 (European Alps), multiply by 0.97 . 10-3 km.we.yr-1 (0.87 . 10-3 / 0.9 ice density = 0.97 . 10-3), it gives2.029 km3.we.yr-1 of mass loss per year. Thus, over 20 years, it results to 40.585 km3 glacier mass loss and finally only over Switzerland (60% of the Alps ice volume), it gives 24.35 km3 glacier mass lost during the last 20 years which is also very close to what you found ! Are my first order calculation right ?

**[AR 1.08]** The reviewer's calculations are correct and are well received. The agreement with the results by Zemp et al. (2019) is not surprising, though, since we used the base data (geodetic mass balance) of that assessment for calibrating our model. We therefore prefer not to add this information in the manuscript as the conclusion might be circular. A

comparison with the results from Zekollari et al. (2019), which is in line with other simulations from Marzeion et al. (2020) as indicated in AR1.04, is now performed (see AR1.09 for more details).

**[RC 1.09]** line 88 to 91: you do not discuss that large part of glacier mass in the Alps is already committed to melt because actual temperatures are already largely higher than pre-industrial values ?

**[AR 1.09]** We added this information:

ll. 99-102: *'Note that, especially for the +1.0°C temperature target, the projected volume and area losses are only slightly higher than the committed loss by 2100 (obtained by applying the 1988-2017 climatic conditions), which projects a loss of 37 ± 6 % and 35 ± 7 % , respectively (for detailed analysis on committed mass loss on multi-century time scales, refer to Zekollari et al. 2019).'*

**[RC 1.10]** line 92 to 93: authors explain the different remaining glaciers in 2100 for different global climate warmings. I wonder how many glacier are committed to disappear in the European Alps under actual climate conditions (that is close to 1.0°C above pre- industrial values) ?

**[AR 1.10]** we addressed this in AR 1.09.

**[RC 1.11]** line 97: do not forget that global average warming of 1.0°C does not result regionally of 1.0°C exactly of warming and thus glaciers can experienced large differences.

**[AR 1.11]** we added this information:
ll.107-109: *'Our results confirm that global average warming can be significantly amplified at the regional scale, and that glaciers in the European Alps sensitively respond to even small levels of global atmospheric warming'*

**[RC 1.12]** line 101 to 113: is it feasible and physically consistent/interesting to go to daily resolution for runoff calculation ?

**[AR 1.12]** Not with our model, unfortunately. For reasons of computational efficiency, our model works with monthly data. For daily runoff calculation, more specific models would be required.

**[RC 1.13]** line 122: I think authors could be more incisive about the irreversible glacier trend. Physically, glaciers growths and retreats are totally consistent and reversible, thus that's not surprising that if global temperature starts to stabilize and reduce after 2100 horizon, glaciers start to grow again at regional scale.

**[AR 1.13]** We fully agree with the reviewer's comment and have amended the paragraph as follows:

*ll.132-140: 'These results show that, owing to slow lowering of air temperatures and enhanced precipitation implied by this particular scenario after 2100, slow glacier recovery might happen (Fig. S3). While glacier volume losses of 80-85% are calculated for 2100, the experiment projects Alpine glaciers to re-gain a total volume that is between 28% and 53% of the 2020 level by 2300. Although this result is only based on three GCM members and is thus very uncertain, it suggests that considering projections beyond 2100 might change the current perception of a possibly irreversible glacier loss. From the physical point of view, the result that glaciers might re-grow after a potential cooling global temperatures is not surprising. Still, increasing the number of GCM members that consider such longer-term horizons and having different research groups performing similar analyses would help verifying the robustness of this preliminary finding. We also stress that decisive climate action would be required for steering global temperatures towards such an evolution (i.e. SSP126).'*

**[RC 1.14]** I am surprised that what seems to me the key message, i.e. 'every half-degree count', is not more highlighting in the paper (in the abstract for example).

**[AR 1.14]** We understand the reviewer's surprise and tried to amend the abstract. Please note that space availability for the abstract in the Brief Communication format is limited to 100 words, and that we have attempted to accommodate RC 2.02 as well:
*'Our results show that even half-degree differences in global temperature targets have important implications for the changes predicted until 2100, and that – for the most optimistic scenarios – glaciers might start to partially recover owing to possibly decreasing temperatures after the end of the 21st century.'*

**[RC 1.15]** Figure 1: to be consistent between panel a) and b), the vertical axis should be either in unit or in unit change (%) for both panels.

**[AR 1.15]** We have added a secondary axis to panel 1b, which now provides both units and units of changes. Expressing a temperature change in % would be very uncommon, and also difficult to interpret.

**[RC 1.16]** Figure 1 and 2: is it consistent to choose a moving average window of 30 years for climate averaging and 20 years for runoff averaging ?

**[AR 1.16]** We thank the reviewer for having noted this inconsistency. We now use a running mean of 30 years in all figures and panels.

**[RC 1.17]** figure 2: why there is no peak water in the curves for annual glacier runoff (panel c) ? How does it compare to other studies ? Does it mean that the peak water is already reached for glaciers in the European Alps ?

**[AR 1.17]** Yes, the reviewer's interpretation is correct. This finding is consistent with earlier studies (e.g. Bliss et al., 2014; Huss and Hock ,2018) and we now acknowledge that in the text with the following sentence:
*ll.115-117: 'The decrease follows above-average glacier melt during the last decades, and indicates that peak glacier discharge for many glaciers of the European Alps already occurred in the past (consistent with the findings of e.g. Huss and Hock, 2018).'*

**[RC 1.18]** supplementary material, table 1: read again the legend which is I think not clear. Probably remove 'given as Area ??' and please explicit 'w.r.t'.

**[AR 1.18]** We reformulated the table caption into:
*'Table S1: Overview of glacier volume change between 2020 and 2100 for glaciers with an area >10 km2. The provided glacier area is from the RGI v6.0.'*

**2. REFEREE'S COMMENTS**

**[RC 2.01]** The paper is clearly written, and the conclusions are clear, some minor comments for improvements are suggested below.

**[AR 2.01]** We thank the reviewer for the very positive feedback.

Specific comments:
**[RC 2.02]** The abstract is very brief and only hints at the results and conclusions. Suggest to include the quantification presented in the conclusion also in the abstract and clarify what 'glaciers might start recovering' actually mean, when does the recovery start (same for all scenarios)? What does recovery mean (full, partial)? Why do they recover? The abstract should really entice the reader to read on so in my opinion more information already here would be useful.

**[AR 2.02]** We thank the reviewer for these important questions but note the 100-words limit imposed by the Brief Communication format only leaves us with marginal room for answering. In the hope that the journal will allow for some flexibility, we propose the following amendment (ll.4-7):
*'Our results show that even half-degree differences in global temperature targets have important implications for the changes predicted until 2100, and that – for the most optimistic scenarios – glaciers might start to partially recover owing to possibly decreasing temperatures after the end of the 21st century.'*

Technical corrections:

**[RC 2.03]** Page 1, Abstract, Line 5, suggest to edit 'temperature targets' with 'scenario' or 'projections resulting in different temperature change'. Suggest also to clarify what 'implications' and what 'changes' are meant, by adding a little more text this sentence would be more informative.

**[AR 2.03]** We changed the sentence following reviewer's suggestion (see AR 2.02)

**[RC 2.04]** Page 1, line 8, suggest to delete 'need to'

**[AR 2.04]** Done

**[RC 2.05]** Page 1, line 12, sentence is not clear, what is ambitious about the targets? What important environmental change is to occur? My suggestion would be to write out what specifically is meant hear.

**[AR 2.05]** We reworded the sentence (ll.12-13) into:
*'Even under these ambitious climate targets, important environmental changes, such as changes in water availability, migration of species, or glacier loss, are expected to occur'.*

**[RC 2.06]** Page 1, line 21-22, suggest to clarify what 'integrated response of climate forcing over decades to centuries' means here. Why is it integrated? What is the time scale? Why decades to centuries?

**[AR 2.06]** We reworded the sentence into:
*'Across the world, glaciers are amongst the most prominent indicators for climatic change, providing visual evidence for climatic changes occurring over decades'*

**[RC 2.07]** Page 2, line 26, suggest to clarify what 'tease out' means and how the authors 'do so'
**[AR 2.07]** We reworded the sentence into:

*ll. 26-29: ' Whilst future projections for the glacier evolution of the European Alps already exist under different representative concentrations pathways (RCPs) (e.g. Zekollari et al., 2019, Marzeion et al., 2020), targeted information on policy-relevant climate targets (like the difference between 1.5 and 2.0°C IPCC, 2018) is difficult to identify. [...]'*

**[RC 2.08]** Page 2, line 53, would be helpful to state what the 0.1° resolution is in km. It it further not clear how the climate is downscaled to the glacier scale, some explanation or statement of how the mass balance (at one point or several) for each glacier is computed.

**[AR 2.08]** We added in the text that 0.1° resolution is 11 km. We acknowledge that the limited space provided within a Brief Communication does not leave room for detailed questions related to the methodology. We have now clarified more explicitly that the methodological steps addressed by the reviewer are described in Huss and Hock (2015).
ll. 62-65: *'In a nutshell, the procedure uses a set of additive and multiplicative correction factors to adjust both the long-term mean and the long-term variability of the coarse-resolution GCMs (100 km) to the level of the high-resolution E-OBS data (see Huss and Hock, 2015, for more details). '*

**[RC 2.09]** Page 3 line 1, here would also be useful to state that the 'coarse' resolution is in km

**[AR 2.09]** We added in the text that it is 100 km.

**[RC 2.10]** Page 3, line 65-66 suggest to turn sentence around, it would be the modelled loss that is close to the observations, rather than the other way

**[AR 2.10]** We turn the sentence around, as suggested by the reviewer.
ll. 72-73:'[…] *our modelled loss is of 24.9 km$^3$, which is very close to the observation-based estimate of 23.6 km$^3$(Grab et al., under review). '*

**[RC 2.11]** Page 3, line 68, suggest to add 'global' between 'century' and 'warming', also would be useful to tell which scenarios those are (it is given a few lines below, my suggestion is to move that information to this location)

**[AR 2.11]** Done

**[RC 2.12]** Page 3 line 75, is it also averaged? How are the grid cells used to produce SMB for each glacier?

**[AR 2.12]** We are not sure to fully understand the reviewer's question. The values of temperature and precipitation change that we report in Figure 1, for example, are indeed averaged over the considered domain (as acknowledged by the figure's caption). For forcing, GloGEMflow's surface mass balance module, instead, the climate information of every grid-cell is considered individually and 'downscaled' for each individual glacier following the procedures described in Huss and Hock (2015). Since this information is now passed at Lines 62-65 in reply to RC 2.08, we do not repeat it here.

**[RC 2.13]** Page 3, line 80 it is not stronger for all three, only for the higher two, the first is decreasing from 0.98 to 0.96, suggest to edit the sentence

**[AR 2.13]** we changed the sentence into:
*'During the summer months (JJA), the temperature increase in the Alps for the two warmer climate targets is even stronger […]'.*

**[RC 2.14]** Page 5, line 107, suggest to edit/replace 'results anticipate' with 'simulations project'

**[AR 2.14]** Done.

**[RC 2.15]** Page 5, line 109, suggest to edit 'is well documented' with something like 'projected in other studies' or 'established'

**[AR 2.15]** We changed 'is well documented' into 'projected in other studies'.

**[RC 2.15]** Page 6, line 115, not clear whether the three GCM members are same as in the previous simulations, 're-run' indicates that, but it could be clarified. If the are same then 'extend' would be clearer. Maybe this information could be added?

**[AR 2.15]** One GCM member (MRI-ESM2-0) is the same, the other two (IPSL-CM6A-LR and CanESM5) were not used for the previous simulations.
We added this information in the manuscript:

ll.127-129: *'To gain insights into glacier evolution beyond this horizon, we run GloGEMflow*

*with three GCM members (one of which was already considered in the 2100 simulations, see Fig. S3) that provide climate data until 2300.'*

**[RC 2.16]** Page 6, line 120 suggest to edit 're-gain a total volume that is between 47% and 72% of the 2020 level', the regained volume is the other part of the 2020 level (53% and 28%), so the sentence in not clear, can it be made clearer? How much is regained?

**[AR 2.16]** We agree the sentence was not clear, and we now reformulated it into:
*'While glacier volume losses of 80-85% are calculated for 2100, the experiment projects Alpine glaciers to re-gain part of the lost volume, reaching a total volume between 28% and 53 % of the 2020 level by 2300. '*
In other words: by 2300, the glaciers are projected to re-gain a volume that is between 1.5 and 3.5 times larger than the one projected for 2100.

**[RC 2.17]** Page 6, line 122, it is not clear what 'perception of an irreversible trend' is, maybe that could be stated, is the perception that the mass loss is irreversible? Where would that perception come from?

**[AR 2.17]** We addressed this already in AR 1.13.
Here we provide a copy of the new paragraph:
*'These results show that, owing to slow lowering of air temperatures and enhanced precipitation implied by this particular scenario after 2100, slow glacier recovery might happen (Fig. S3). While glacier volume losses of 80-85% are calculated for 2100, the experiment projects Alpine glaciers to re-gain a total volume that is between 28% and 53% of the 2020 level by 2300. Although this result is only based on three GCM members and is thus very uncertain, it suggests that considering projections beyond 2100 might change the current perception of a possibly irreversible glacier loss. From the physical point of view, the result that glaciers might re-grow after a potential cooling global temperatures is not surprising. Still, increasing the number of GCM members that consider such longer-term horizons and having different research groups performing similar analyses would help verifying the robustness of this preliminary finding. We also stress that decisive climate action would be required for steering global temperatures towards such an evolution (i.e. SSP126).'*

**[RC 2.18]** Page 6, line 123, suggest to replace 'verifying' with 'verify'

**[AR 2.18]** Done

**[RC 2.19]** Page 7 line 125, this sentence is not clear, what is 'decisive acting'? what are 'unwanted consequences'? where is the 'overwhelming consensus'? suggest to turn sentence around the temperature target (rather than 'climate target')

**[AR 2.19]** The wording 'decisive acting' was unfortunate, and we now make reference to some of the most important reports for clarifying the other two questions. The wording 'temperature target' was adopted as suggested. The revised sentence reads:

ll.142-145:*' Whilst there is overwhelming consensus that decisive action has to be taken to limit unwanted consequences of ongoing climatic change* (UN, 2015; IPCC, 2018; IPBES, 2019; IPCC, 2019; WEF, 2020) *, the debate around which temperature targets to pursue is*

*all but settled'*

**[RC 2.20]** Page 7, line 128, suggest to replace 'showed' with 'show'

**[AR 2.20]** Done

**[RC 2.21]** Page 7 line 131, suggest to replace 'would' with 'will'

**[AR 2.21]** Done

**[RC 2.22]** Page 7, line 135, edit sentence, it is not the changes, but rather the peak runoff that occurs 1 to 2 months earlier. Suggest also to replace 'anticipated' with 'projected'

**[AR 2.22]** We edited the sentence into:
ll.154-155: *'Changes in monthly runoff – with a runoff peak projected to occur 1 to 2 months earlier by the end of the century -- will be even more pronounced '*

**[RC 2.23]** Page 7 line 136, suggest to add 'peak' between 'August' and 'runoff'

**[AR 2.23]** Done

**[RC 2.24]** Figure 2. Why is there a bed upwards (kink) for +2°C (red line) at 2020?, less for the +1.5°C (light blue line) and downward dip that goes up for the +1°C (blue line) is this due to the transition from the E-OBS to CMIP6 models? This could be discussed in text. Caption, line 3, replace 'or' with 'of' and suggest to add that the (n) is given in panel (a).

**[AR 2.24]** The upwards kink for the +2°C was an artifact introduced by the running mean. This is now corrected, also in response to request RC 1.16. The caption was amended as suggested and now reads:
'Figure 2. *Modelled evolution of total glacier (a) volume, (b) area, (c) annual glacier runoff, and (d) monthly glacier runoff of the European Alps. Time series in c are smoothed with a 30-year running mean. In all panels, the thick line represents the mean and the transparent band corresponds to one standard deviation of the results obtained by forcing GloGEMflow with the selected GCM members. The numbers of GCM members is given (n) in panel (a). '*

**Supplementary material**

**[RC 2.25]** Figure S2, Delete 'annual' in figure title after 'winter'

**[AR 2.25]** Done

**[RC 2.26]** In figure captions of S1 and S2 suggest to replace 'of 72 glaciers' with 'from 72 glaciers'

**[AR 2.26]** Done

**[RC 2.27]** Suggest to edit figure caption S3 it is not only Modelled glacier evolution until 2300 but also temperature and precipitation evolution.

**[AR 2.27]** We changed it into 'Evolution until 2300'

**[RC 2.28]** Table S2.1 replace 'and' with 'an' before 'area'
something strange in the parenthesis what does (given as 'Area??) refer to?

**[AR 2.28]** We reformulated the caption into: *'Table S1: Overview of glacier volume change between 2020 and 2100 for glaciers with an area >10 km2. The provided glacier area is from the RGI v6.0.'*

REFERENCES:

Bliss, A., Hock, R., and Radić, V. (2014), Global response of glacier runoff to twenty-first century climate change, *J. Geophys. Res. Earth Surf.*, 119, 717– 730, doi:10.1002/2013JF002931.

Compagno, L., Zekollari, H., Huss, M., & Farinotti, D. (2021). Limited impact of climate forcing products on future glacier evolution in Scandinavia and Iceland. *Journal of Glaciology,* 1-17. doi:10.1017/jog.2021.24

Huss, M., Farinotti D., Bauder, A. and Funk, M. (2008). Modelling runoff from highly glacierized alpine drainage basins in a changing climate. *Hydrological Processes*, 22(19), 3888-3902, doi:10.1002/hyp.7055

---

## Author Response (AR2)

**Author's response to the comments received for tc-2021-31**

The following pages contain a point-by-point reply to the comments provided by the two referees that reviewed our first submission (TC-2021-31)

Each of the editor's (**EC**) and referee's comment (**RC**) is numbered. If a comment contained several points, we numbered them, and address them individually in our author replies (**AR**).

We also carefully re-checked the formulation of the entire manuscript, and amended the text where appropriate."

**1. EDITOR'S COMMENT**

**[EC 1.01]** line 24: does this relation between CO2 kg and loss of glacier mass apply at the global scale or European Alps scale? This should be specified.

**[AR 1.01]** It applies at the global scale. We added this information in the manuscript:
ll. 23-24: '*Previous work estimated that every kg of additionally emitted $CO_2$ would result in a long-term global glacier mass loss of ca. 16 kg (Marzeion et al., 2018)*'

**[EC 1.02]** line 130: if all three GCM members are giving T < +2°C by 2100, why only one was considered in the 2100 analysis? From line 80, I understand that you took all available GCM members giving temperature between +075 and 2.25°C at 2100? Why these two members were excluded for the 2100 analysis?

**[AR 1.02]** This information was not clear. Of the three GCM members which go until 2300, only one has a T < +2°C by 2100, whilst the remaining two have a T > +2°C by 2100, but have a T < +2°C by 2300, i.e. the temperature after 2100 decreases. Therefore, the last two GCM members were used only for the simulations which go up to 2300.
We reformulated it in the manuscript as:
*ll. 127-131:'To gain insights into glacier evolution beyond this horizon, we run GloGEMflow with three GCM members that provide climate data until 2300 and that project mean global temperature changes below +2.0°C for 2300 (see Fig. S3). Note that one of these GCM members was already considered in the simulations until 2100, whilst the remaining two were not because they show a warming beyond +2.0°C for 2100 (Fig. S3).'*

**2. REFEREE'S COMMENT**

**[RC 1.01]** Line 39: please consider to remove "in-house"

**[AR 1.01]:** Done

**[RC 1.02]** Line 61: please consider to add something like " at least regionally for European glaciers " or something similar at the end of the sentence.

**[AR 1.02]** Please note that the study cited in the sentence mentioned by the reviewer is not specific to the European Alps. The suggested addition would thus be misleading and we therefore decided to leave the sentence unchanged.